# Resource availability and barriers to delivering quality care for newborns in hospitals in the southern region of Malawi: A multisite observational study

**Mtisunge Joshua Gondwe**[1,2]*, **Nicola Desmond**[2,3], **Mamuda Aminu**[3], **Stephen Allen**[1]

**1** Department of Clinical Sciences, Liverpool School of Tropical Medicine, Liverpool, United Kingdom, **2** Behaviour and Health Group, Malawi Liverpool Wellcome Trust- Clinical Research Programme, Blantyre, Malawi, **3** Department of International Public Health, Liverpool School of Tropical Medicine, Liverpool, United Kingdom

* mtisungejoshua@gmail.com

## Abstract

Facility-based births have increased in low and middle-income countries, but babies still die due to poor care. Improving care leads to better newborn outcomes. However, data are lacking on how well facilities are prepared to support. We assessed the availability of human and material resources and barriers to delivering quality care for newborns and barriers to delivering quality care for newborns. We adapted the WHO Service Availability and Readiness Assessment tool to evaluate the resources for delivery and newborn care and barriers to delivering care, in a survey of seven hospitals in southern Malawi between January and February 2020. Data entered into a Microsoft Access database was exported to IBM SPSS 26 and Microsoft Excel for analysis. All hospitals had nursery wards with at least one staff available 24 hours, a clinical officer trained in paediatrics, at least one ambulance, intravenous cannulae, foetal scopes, weighing scales, aminophylline tablets and some basic laboratory tests. However, resources lacking some or all of the time included anticonvulsants, antibiotics, vitamin K, 50% dextrose, oxytocin, basic supplies such as cord clamps and nasal gastric tubes, laboratory tests such as bilirubin and blood culture and newborn clinical management guidelines. Staff reported that the main barriers to providing high-quality care were erratic supplies of power and water, inadequacies in the number of beds/cots, ambulances, drugs and supplies, essential laboratory tests, absence of newborn clinical protocols, and inadequate staff, including paediatric specialists, in-service training, and support from the management team. In hospitals in Malawi, quality care for deliveries and newborns was compromised by inadequacies in many human and material resources. Addressing these deficiencies would be expected to lead to better newborn outcomes.

## Introduction

Globally, every minute nearly four newborn babies are stillborn and five die during the neonatal period (the first 28 days of life) [1, 2]. About 80% of stillbirths and neonatal deaths occur in

**Data Availability Statement:** All data relevant to the study are included in the article or uploaded as supplementary information.

**Funding:** MJG received a Commonwealth Commission Doctoral Scholarship (MWCSC, 2018, 802). MJG was also supported by a Wellcome Strategic award number 206545/Z/17/Z that supports Malawi Liverpool Wellcome Trust Clinical Research Programme (MLW). The funders had no role in study design, data collection and analysis, decision to publish, or preparation of the manuscript.

**Competing interests:** The authors have declared that no competing interests exist.

low-middle income countries (LMICs) with sub-Saharan Africa accounting for 42% of these deaths.

A third of stillbirths occur during labour and on the day of birth [3, 4]. Globally in 2019, neonatal deaths contributed about 47% of all under-five deaths. About a third of all neonatal deaths occur within the first day of life and three quarters within the first week [5]. Malawi contributes significantly to global mortality, with perinatal and neonatal mortality rates at 35 per 1000 total births and 27 per 1000 live births in 2016, respectively [6]. To achieve the Sustainable Development Goal 3.2 and the vision and goals of the Every Newborn: an action plan to end preventable deaths [7], a focus on reducing neonatal mortality, especially during birth and the first day and week of life is crucial.

Most stillbirths and neonatal deaths result from preventable causes and are associated with poor quality of care during and after birth [1, 2, 8–10]. Good quality care could prevent almost two neonatal deaths every minute [9]. Providing quality care benefits both patient outcomes [6] and the health system [11].

Birth asphyxia (48%), complications of preterm delivery (40%) and sepsis (5%) have been identified as the leading causes of neonatal deaths in Malawi [12]. Improving care during the antepartum period, labour, delivery and postpartum could prevent stillbirths and neonatal deaths [13]. But the quality of care delivered by the facilities in Malawi falls significantly short of global standards of evidence-based care, despite nine out of ten women attending antenatal care and delivering at facility. Increased facility births has increased workload in many hospitals [10] and lack of staff and equipment compromises the quality of care in most LMICs especially for vulnerable groups [9, 11]. Delay in the decision to deliver has been identified as one of the contributing factor to newborn deaths. Although there has been a slight increase in the proportion of caesarian deliveries in Malawi, from 5% (2010) to 6% (2016) [6], the average decision to incision time of 1.69 hours is still too long to save the life of the baby during complicated pregnancy [14]. Furthermore only 60% of newborns in Malawi receive postnatal checks within 48 hours of delivery which is key for early identification of complications and prompt care [6]. At the community level, challenges such as cultural beliefs that prevent care seeking for sick newborns, particularly during the first seven days of life, and beliefs that do not value and honour the life of a newborn, especially low birth weight or preterm babies, also impact on mortality. To provide a good standard of care and a better user experience, health systems must be well prepared with material resources and sufficient staff with the knowledge, skills and capacity to deal with normal and complicated pregnancies, childbirths and the newborn [15].

Previous studies that assessed the quality of maternal or newborn care in Malawian health facilities highlighted challenges with infrastructure, transportation, communication, staff training, treatment guidelines and material and human resources [5, 16, 17]. It is important to assess additional aspects that affect the delivery of newborn care but were not covered in the earlier studies, such as staff availability in wards and leadership and governance challenges. It is also crucial to assess resource availability to support the application of the WHO's quality of care standards guidelines, which have been locally adapted for use in Malawi but not discussed in earlier studies.

Assessing quality of health care using recognised standards, criteria and indicators is key to quality care improvement [18]. WHO published a framework for improving the quality of care for mothers and newborns [19]. Building on these frameworks and covering two of WHO's six vision strategic areas of standards of care and measures of quality of care [19], WHO has published recently standards for improving the care of small and sick newborns [20]. The standards are guided by the eight domains of the WHO quality care framework [15] that health facility leaders, planners, managers and providers can use to assess and monitor the availability

of resources, performance, areas for improvement and the impact of the intervention [19]. A standard is what is expected to be provided to ensure high-quality care [20]. Quality statements explain how to achieve the standard of care and are accompanied by quality measures used to assess, measure, and monitor inputs, processes, outputs and outcomes [20]. The eight standards are summarised in the Supporting Information (S1 Table) and address the following areas: practice based on research (evidence base practice); Information system that can be used (actionable information system); dependable system of referral (functional referral system); effective communication and family participation; respective women and newborn care; care that is supportive emotionally, developmentally and psychologically; multidisciplinary staff that are skilled, motivated and compassionate and physical resources (drugs, supplies, infrastructure, and equipment) that are necessary for pregnant women, babies and sick and small babies [20].

Malawi adapted all eight WHO quality of care framework standards and added a ninth standard regarding community health care and social accountability in maternal and neonatal health and published these for use in April 2020 [21]. However, Malawi has not yet adapted recently published WHO standards for improving small and sick newborn [20].

Little is known on resource availability in LMICs to meet the quality-of-care standards set by WHO [19, 20, 22]. This study was conducted between January to February 2020 in the context of a larger study evaluating stillbirth and neonatal death audit in the southern region of Malawi We assessed the availability of human and material resources and barriers to delivering quality care for newborns.

## Materials and methods

### Study design and setting

This survey was done in seven public hospitals in the southern region of Malawi purposively selected to represent health facilities in the region with neonatal mortality at district level ranging from 15 to 30 per 1000 live births [6]. The selected hospitals included one central hospital (tertiary level, hospital 1) and six district hospitals (secondary level, hospitals 2–7). The central hospital provides specialised inpatient and outpatient care at a regional level and serves referrals from the district hospitals and health centres within the region. The district hospitals provides both outpatient and inpatient services and serves referrals from community hospitals and health centres. Health centres were not included as they do not admit neonates.

### Data collection tools

We adapted the WHO Service Availability and Readiness Assessment (SARA) tool [23] to assess facility readiness to provide quality newborn care during birth and up to 28 days postnatally. Information was collected regarding facility characteristics, infrastructure, transport and communication, staff availability in labour, postnatal and neonatal wards, staff training in the last 12 months, material inventories (essential supplies, drugs, equipment and laboratory), clinical protocols, leadership and governance (S1 Appendix). We piloted the tool in hospital 2 and incorporated appropriate changes such as including pharmacy stock out days to better achieve a comprehensive view of care. The data collection methods included observed availability in a service area, interviews with wards in charge and laboratory and pharmacy staff and stock card checking.

### Administering the health facility assessment

Initially, the study team introduced the study to central and district management teams and secured permission to conduct the study. An introductory meeting was then held with facility

management and staff. The first author (MG) conducted the health facility assessments between January to February 2020. The health facility resource survey was carried out with senior or ward nurses in charge of the labour, postnatal and neonatal wards, and pharmacy and laboratory staff available on the assessment day. During interviews, staff were asked if resources were always, sometimes, or never available in the previous three months. "Always available" means the resource was available every day for the last three months (no days of stockout) or the service was provided all of the time at the facility, or the staff cadre was available all of the time during the shift. "Sometimes" means the resource was not available every day throughout the course of the past three months (some stockout days) or the service was occasionally provided, or the staff cadre was occasionally on duty. "Never" means that the resource or supply was not available during the entire three-month period, or a service was never provided at the facility or staff cadre was not on duty at any point during the shift. The health resource survey took two days in each hospital.

### Data analysis

Data extracted from paper forms was entered into a Microsoft Access database. Data were checked for anomalies by running descriptive summaries and data entry errors were corrected. There was no imputation of missing data. Data were exported to IBM SPSS 26 and Microsoft Excel for analysis. Descriptive statistics were used to summarise scores allowing comparisons between hospitals.

### Ethics statement

Approval was obtained from the College of Medicine (P.11/19/2869) and the Liverpool School of Tropical Medicine (19–076) ethics committees. All hospitals gave permission to conduct the study. All healthcare workers who took part in the study signed a written ethics-approved informed consent form.

### Inclusivity in global research

Additional information regarding the ethical, cultural, and scientific considerations specific to inclusivity in global research is included in the Supporting Information (S1 Checklist).

## Results

### Hospital characteristics

All seven hospitals offered maternal and neonatal services 24 hours each day. Each hospital had labour, postnatal and nursery wards/area. All seven hospitals admitted babies up to age 2 months requiring specialised care. Bed capacity and staffing levels for each hospital are shown in Table 1. Although there was a particular shortage of nursing/midwifery officers across all hospitals, overall, there was marked variability between hospitals in the proportion of staff posts filled. This contributed to marked discrepancies between hospitals in the number of staff in post according to the number of beds in all wards. Although none of the hospitals had a full-time paediatrician, all hospitals had at least one general medical officer available for consultation in all wards during both day and night shifts. All hospitals had at least one clinical officer trained in obstetrics and gynaecology or paediatrics, a clinical technician trained at diploma level, at least one registered nurse/midwife trained at degree level and at least three nurse/midwife technicians trained at diploma level and stationed at each ward.

**Table 1. Bed capacity and staffing levels[1,2,3].**

|  | Central hospital | District hospitals | | | | | |
|---|---|---|---|---|---|---|---|
| **Hospital** | **1** | **2** | **3** | **4** | **5** | **6** | **7** |
| **Total deliveries per year** | **7302** | **7511** | **4659** | **4143** | **10898** | **4486** | **6411** |
| **Total neonatal admissions per year** | **2734** | **1778** | **1283** | **744** | **1902** | **809** | **1463** |
| Number of Labour ward beds | 9 | 8 | 8 | 7 | 8 | 4 | 9 |
| Number of postnatal beds | 31 | 88 | 54 | 18 | 38 | 35 | 33 |
| Number of nursery beds | 52 | 41 | 26 | 10 | 12 | 22 | 20 |
| **Number of Staff Labour ward** |  |  |  |  |  |  |  |
| Medical Officer/Specialists | 2 | 1 | 1 | 2 | 3 | 1 | 2 |
| Clinical Officer/Technician | 1 | 1 | 2 | 3 | 1 | 3 | 4 |
| Nursing/Midwifery Officers | 7 | 1 | 5 | 4 | 2 | 2 | 3 |
| Nurse/midwife technician | 11 | 5 | 9 | 6 | 9 | 5 | 11 |
| Support staff | 9 | 5 | 5 | 7 | 10 | 12 | 12 |
| **Number of Staff Postnatal ward** |  |  |  |  |  |  |  |
| Medical Officer/Specialists | 0 | 0 | 0 | 0 | 0 | _ | 0 |
| Clinical Officer/Technician | 1 | 3 | 2 | 3 | 3 | _ | 3 |
| Nursing/Midwifery Officers | 6 | 2 | 2 | 2 | 2 | _ | 2 |
| Nurse/midwife Technician | 9 | 4 | 4 | 5 | 10 | _ | 10 |
| Support staff | 9 | 5 | 7 | 6 | 12 | _ | 12 |
| **Number of staff nursery ward** |  |  |  |  |  |  |  |
| Medical Officer/Specialists | 0 | 0 | 0 | 0 | 0 | 0 | 0 |
| Clinical Officer/Technician | 2 | 1 | 2 | 3 | 5 | 1 | 1 |
| Nursing/Midwifery Officers | 6 | 1 | 2 | 2 | 1 | 3 | 2 |
| Nurse/midwife technician | 8 | 6 | 4 | 3 | 5 | 1 | 5 |
| Support staff | 6 | 6 | 1 | 0 | 0 | 0 | 0 |

[1] One month data for hospital 3- and two-months data for hospital 4 were missing

[2] Dash (-) postnatal ward within labour ward

[3] Nursing/Midwifery Officer trained at degree level while nurse/midwife technician trained at diploma level. Clinical Officers are trained at degree level while clinical technicians are trained at diploma level. Support staff includes patients and hospital attendants

## Infrastructure

Inadequate electricity and water supply were noted in all hospitals with an adequate power backup system in hospital 3 only. Water supply failed during power cuts, but all hospitals had reservoir tanks and buckets with a tap for handwashing when piped water was unavailable.

## Transport and communication

Although three hospitals were allocated one or more ambulances per 50,000 population, only about two-thirds of ambulances were functional so that only hospital 6 had adequate provision (S2 Table). All hospitals had always one functional landline telephone or mobile phone for use.

## Staff availability and training

Table 1 summarises the number of staff allocated in the wards, while Table 2 summarises the availability of staff during day and night shift in the wards. Nurses/midwife technicians and support staff were the only staff always available during day and night shifts in all wards. One clinical officer or technician was available on call during the night shift. Twenty five percent or

**Table 2. Staff availability[1,2].**

**Labour ward**

| Facility | Availability day | | | | | Availability night | | | | |
|---|---|---|---|---|---|---|---|---|---|---|
| | Medical Officer | Registered nurse/ Midwife | Clinical Officer | Nurse/Midwife Technician | Support Staff | Medical Officer | Registered nurse/ Midwife | Clinical Officer | Nurse/Midwife Technician | Support Staff |
| Hospital 1 | Yellow | Green | Green | Green | Green | Yellow | Yellow | Green | Green | Green |
| Hospital 2 | Yellow | Yellow | Green | Green | Green | Yellow | Red | Yellow | Green | Green |
| Hospital 3 | Yellow | Yellow | Yellow | Green | Green | Yellow | Yellow | Yellow | Green | Green |
| Hospital 4 | Yellow | Yellow | Yellow | Green | Green | Red | Yellow | Yellow | Green | Green |
| Hospital 5 | Yellow | Green | Green | Green | Green | Red | Yellow | Yellow | Green | Green |
| Hospital 6 | Yellow | Green | Green | Green | Green | Yellow | Yellow | Green | Green | Green |
| Hospital 7 | Yellow | Green | Green | Green | Green | Yellow | Yellow | Yellow | Green | Green |

**Postnatal ward**

| Facility | Availability day | | | | | Availability night | | | | |
|---|---|---|---|---|---|---|---|---|---|---|
| | Medical Officer | Registered nurse/ Midwife | Clinical Officer | Nurse/Midwife Technician | Support Staff | Medical Officer | Registered nurse/ Midwife | Clinical Officer | Nurse/Midwife Technician | Support Staff |
| Hospital 1 | Yellow | Green | Green | Green | Green | Yellow | Yellow | Yellow | Green | Green |
| Hospital 2 | Yellow | Green | Green | Green | Green | Yellow | Yellow | Yellow | Green | Green |
| Hospital 3 | Yellow | Green | Green | Green | Green | Yellow | Yellow | Green | Green | Green |
| Hospital 4 | Yellow | Green | Green | Green | Green | Red | Yellow | Yellow | Green | Green |
| Hospital 5 | Yellow | Green | Green | Green | Green | Red | Red | Green | Green | Green |
| Hospital 6 | Yellow | Green | Green | Green | Green | Yellow | Yellow | Yellow | Green | Green |
| Hospital 7 | Yellow | Green | Green | Green | Green | Yellow | Yellow | Yellow | Green | Green |

**Nursery ward**

| Facility | Availability day | | | | | Availability night | | | | |
|---|---|---|---|---|---|---|---|---|---|---|
| | Medical Officer | Registered nurse/ Midwife | Clinical Officer | Nurse/Midwife Technician | Support Staff | Medical Officer | Registered nurse/ Midwife | Clinical Officer | Nurse/Midwife Technician | Support Staff |
| Hospital 1 | Green | Green | Yellow | Green | Green | Yellow | Yellow | Yellow | Green | Green |
| Hospital 2 | Yellow | Yellow | Green | Green | Green | Yellow | Red | Yellow | Green | Green |
| Hospital 3 | Yellow | Yellow | Yellow | Green | Green | Yellow | Yellow | Yellow | Green | Green |
| Hospital 4 | Yellow | Yellow | Green | Green | Green | Red | Yellow | Yellow | Green | Green |
| Hospital 5 | Yellow | Yellow | Yellow | Green | Green | Red | Yellow | Yellow | Green | Green |
| Hospital 6 | Yellow | Yellow | Green | Green | Green | Yellow | Yellow | Yellow | Green | Green |
| Hospital 7 | Yellow | Yellow | Yellow | Green | Green | Yellow | Yellow | Yellow | Green | Green |

[1] Green = Always, Yellow = Sometimes, Red = Never

[2] **Always**- staff cadre available all the time during the shift. **Sometimes**- Occasionally on duty. **Never**—Not on duty at any point during the shift

less of ward staff were trained in each category assessed despite a recommendation that all ward members should be trained or refreshed every 12 months [19]. In the previous 12 months, only 22.7% of clinicians and nurses across all hospitals had been trained in Integrated Maternal and Neonatal Care (IMNC), 23.1% in Helping Babies Breathe (HBB), 15.6% in Care of Infant and Newborn (COIN), and 25.3% in maternal and neonatal death audit (S3 Table).

## Essential supplies, drugs and equipment

All hospitals had at least one essential drug or supply out of stock in the month preceding the assessment (S4 Table). The number of stockout days, however, varied amongst the facilities and ranged from 1 to 9 essential supplies out of stock. For instance, hospital 2 had four essential supplies (cord clamp, nasogastric tubes, nasal prongs, and thermometers) out of stock for 31 days as well as four essential drugs (50 percent dextrose, gentamycin, ceftriaxone, and vitamin K) out of stock for two to thirty days in the month prior to the assessment, whereas hospital 1 only had cord clamps out of stock for 23 days, in the same period. Even though some supplies and drugs were available at the pharmacy, their availability in wards varied, with most drugs not always available especially 50% dextrose, diazepam, magnesium sulphate, benzylpenicillin, gentamycin, ceftriaxone and Vitamin K (Table 3). Despite glucometer and blood pressure machines often being available, there were challenges is supplies of glucose test strips and blood pressure batteries for them to function. Basic supplies such as cord clamps, nasogastric tubes and urine dipsticks were not always available in hospitals. Labour and nursery wards had at least one to two functional essential equipment oxygen concentrator, sunction machine, bag and mask and resuscitaire always available, but postnatal wards lacked such essential equipment for the care of the neonate. Only nursery wards were equipped with CPAP and photo-therapy lamps for the management of newborns with respiratory distress and jaundice, respectively.

## Laboratory tests

Basic laboratory diagnostic tests were provided except for haematocrit, bilirubin, blood gas analysis and blood cultures that were uniformly not provided. (Table 4).

## Clinical protocols

Clinical protocols for neonatal resuscitation, care of small and preterm babies, care of the sick neonate and essential newborn care were available in some of the nursery wards but not in the labour and postnatal wards (Table 5). Infection prevention protocols were absent in 9/21 (43%) wards assessed. Protocols for the management of complications of labour were available in almost all labour and postnatal wards.

## Leadership and governance

Neonatal outcome data were summarised and pasted on the wall in wards of hospitals 1, 3, 5 and 7. Staff appraisals had been performed in the last 12 months only in hospital 2. Supportive supervision was done by national-level (Ministry of Health) staff but not by the District Health Management Team (DHMT) in all facilities. All seven facilities reported having functional neonatal death audit teams in place though the frequency of neonatal death audit meetings differed, with only 2 hospitals reported having audits in the month prior to the survey. Quality improvement teams were available but inactive in all facilities.

**Table 3. Availability of essential supplies, drugs and equipment[1,2,3].**

| RESOURCES | Hospital 1 | | | Hospital 2 | | | Hospital 3 | | | Hospital 4 | | | Hospital 5 | | | Hospital 6 | | | Hospital 7 | | |
|---|---|---|---|---|---|---|---|---|---|---|---|---|---|---|---|---|---|---|---|---|---|
| | Labour ward | Postnatal ward | Neonatal ward | Labour ward | Postnatal ward | Neonatal ward | Labour ward | Postnatal ward | Neonatal ward | Labour ward | Postnatal ward | Neonatal ward | Labour ward | Postnatal ward | Neonatal ward | Labour ward | Postnatal ward | Neonatal ward | Labour ward | Postnatal ward | Neonatal ward |
| **ESSENTIAL SUPPLIES** | | | | | | | | | | | | | | | | | | | | | |
| Intravenous cannula | | | | | | | | | | | | | | | | | | | | | |
| Intravenous fluids | | | | | | | | | | | | | | | | | | | | | |
| Giving sets | | | | | | | | | | | | | | | | | | | | | |
| Sterile blade | | | | | | | | | | | | | | | | | | | | | |
| Cord clamp | | | | | | | | | | | | | | | | | | | | | |
| Nasal gastric tubes | | | | | | | | | | | | | | | | | | | | | |
| Heaters | | | | | | | | | | | | | | | | | | | | | |
| Nasal prongs | | | | | | | | | | | | | | | | | | | | | |
| Guedel airway | | | | | | | | | | | | | | | | | | | | | |
| Glucometer | | | | | | | | | | | | | | | | | | | | | |
| Glucometer test stripes | | | | | | | | | | | | | | | | | | | | | |
| Thermometers | | | | | | | | | | | | | | | | | | | | | |
| BP apparatus | | | | | | | | | | | | | | | | | | | | | |
| BP Calf batteries | N/U | N/U | N/U | N/U | N/U | N/U | N/U | N/U | N/U | N/U | N/U | N/U | N/U | N/U | N/U | | N/U | N/U | | N/U | N/U |
| Foetal scope | N/U | | N/U | N/U | | N/U | N/U | | N/U | N/U | | N/U | N/U | | N/U | | | N/U | | N/U | N/U |
| weighing scale | | | | | | | | | | | | | | | | | | | | | |
| urine dipsticks | | | | | | | | | | | | | | | | | | | | | |
| **ESSENTIAL DRUGS** | | | | | | | | | | | | | | | | | | | | | |
| 50% dextrose | | | | | | | | | | | | | | | | | | | | | |
| Diazepam | | | | | | | | | | | | | | | | | | | | | |
| Phenobarbitone | | | | | | | | | | | | | | | | | | | | | |
| Magnesium Sulphate | N/U | | N/U | N/U | | N/U | N/U | | N/U | N/U | | N/U | N/U | | N/U | | | N/U | | | N/U |
| Benzylpenicillin | | | | | | | | | | | | | | | | | | | | | |
| Gentamycin | | | | | | | | | | | | | | | | | | | | | |
| Ceftriaxone | | | | | | | | | | | | | | | | | | | | | |
| Oxytocin | NU | | N/U | N/U | | N/U | NU | | NU | N/U | | N/U | N/U | | N/U | | | N/U | | N/U | N/U |
| Dexamethasone | | | | | | | | | | | | | | | | | | | | | |
| Vitamin K | | | | | | | | | | | | | | | | | | | | | |
| Aminophylline tablets | | | | | | | | | | | | | | | | | | | | | |
| Artesunate | | | | | | | | | | | | | | | | | | | | | |
| **ESSENTIAL EQUIPMENT** | | | | | | | | | | | | | | | | | | | | | |
| Bag and Mask | | | | | | | | | | | | | | | | | | | | | |
| Resuscitaire | | | | | | | | | | | | | | | | | | | | | |
| Suction machine | | | | | | | | | | | | | | | | | | | | | |
| Oxygen concentrator | | | | | | | | | | | | | | | | | | | | | |
| CPAP | | | | | | | | | | | | | | | | | | | | | |
| Phototherapy | | | | | | | | | | | | | | | | | | | | | |

[1]Green = Always, Orange = Sometimes, Red = Never

[2]N/U- Equipment is only used in Labour wards not in postnatal and nursery wards

[3]CPAP- Continuous positive airway pressure

**Table 4. Laboratory[1].**

| Resources | Central hospital | District hospitals | | | | | |
|---|---|---|---|---|---|---|---|
| | 1 | 2 | 3 | 4 | 5 | 6 | 7 |
| Full Blood Count | 🟩 | 🟩 | 🟩 | 🟩 | 🟩 | 🟩 | 🟩 |
| Bilirubin | 🟥 | 🟥 | 🟥 | 🟥 | 🟥 | 🟥 | 🟥 |
| Blood glucose | 🟩 | 🟩 | 🟩 | 🟩 | 🟩 | 🟩 | 🟩 |
| Malaria Smear | 🟩 | 🟩 | 🟩 | 🟩 | 🟩 | 🟩 | 🟩 |
| Grouping and Crossmatch | 🟩 | 🟩 | 🟩 | 🟩 | 🟩 | 🟩 | 🟩 |
| CSF analysis | 🟩 | 🟩 | 🟩 | 🟩 | 🟩 | 🟩 | 🟩 |
| Haematocrit (PCV) | 🟥 | 🟥 | 🟥 | 🟥 | 🟥 | 🟥 | 🟥 |
| Haemoglobin | 🟩 | 🟩 | 🟩 | 🟩 | 🟩 | 🟩 | 🟩 |
| Arterial blood gases | 🟥 | 🟥 | 🟥 | 🟥 | 🟥 | 🟥 | 🟥 |
| Urine Microscopy | 🟩 | 🟩 | 🟩 | 🟩 | 🟩 | 🟩 | 🟩 |
| Urine dipstick | 🟩 | 🟩 | 🟩 | 🟩 | 🟩 | 🟩 | 🟩 |
| HIV | 🟩 | 🟩 | 🟩 | 🟩 | 🟩 | 🟩 | 🟩 |
| Syphilis | 🟩 | 🟩 | 🟩 | 🟩 | 🟩 | 🟩 | 🟩 |
| Blood Culture | 🟥 | 🟥 | 🟥 | 🟥 | 🟥 | 🟥 | 🟥 |

[1]Green = Test conducted at hospital, Red = Test not conducted at hospital

## Discussion

### Summary of findings

All hospitals provided maternal and newborn health services and had at least one clinical officer trained in paediatrics and one healthcare worker available 24 hours a day to provide care to sick neonates. All hospitals had a separate nursery ward or unit dedicated to neonatal care though bed capacity varied. All hospitals had clinical protocols for managing labour complications pasted on the walls of the labour and postnatal wards. Some essential supplies and laboratory tests were always available.

However, many essential drugs and basic supplies were not always available for mothers and newborns. Clinical protocols for neonatal resuscitation, infection prevention, care of small and preterm babies, care of the sick neonate and essential newborn care were not available in some hospitals. Staff reported several barriers to providing high-quality care including inadequate beds, erratic power and water supplies, inadequate ambulances, inadequate in-service staff training, unavailability of other staff cadres during the night (as only nursing staff is always available), lack of paediatrician specialists, inadequate drugs, supplies and essential laboratory tests, absence of newborn clinical protocols and inadequate support from management teams. Supporting Information (S1 Table) summarises the key standards relevant to the data captured in this study.

### Infrastructure

Our finding regarding nursery ward space is inconsistent with the WHO 2020 standards for providing high-quality care to small and sick newborns [20]. Limited nursery beds resulted in nursing more than one baby in a single cot increasing the risk of infection and overburdening staff [24–27]. Hospital-acquired infections among neonates cause about 30–40% of neonatal deaths [28]. With more than 90% of women delivering at the facility and inadequate ward infrastructure, the nursery wards are always full, resulting in nursing more than one baby in cot [6].

**Table 5. Availability of clinical protocols in wards[1].**

| Protocols and governance | Central hospital | | | District hospitals | | | | | | | | | | | | | | | | | |
|---|---|---|---|---|---|---|---|---|---|---|---|---|---|---|---|---|---|---|---|---|---|
| | 1 | | | 2 | | | 3 | | | 4 | | | 5 | | | 6 | | | 7 | | |
| | Labour ward | Postnatal ward | Nursery ward | Labour ward | Postnatal ward | Nursery ward | Labour ward | Postnatal ward | Nursery ward | Labour ward | Postnatal ward | Nursery ward | Labour ward | Postnatal ward | Nursery ward | Labour ward | Postnatal ward | Nursery ward | Labour ward | Postnatal ward | Nursery ward |
| **Clinical protocols pasted on the wall** | | | | | | | | | | | | | | | | | | | | | |
| Neonatal resuscitation | Green | Red | Green | Green | Red | Green | Green | Red | Green | Green | Green | Green | Green | Red | Red | Green | Red | Green | Green | Red | Green |
| Infection prevention | Green | Green | Green | Green | Green | Red | Green | Green | Green | Green | Green | Green | Green | Red | Green | Green | Green | Green | Green | Red | Red |
| Care of small and preterm babies | Red | Red | Green | Green | Red | Green | Red | Green | Green | Green | Red | Green | Green | Red | Green | Green | Green | Green | Green | Red | Green |
| Care of the sick neonate | Red | Red | Green | Red | Green | Green | Green | Red | Green | Red | Red | Green | Red | Green | Green | Red | Red | Green | Red | Red | Red |
| Essential newborn care | Green | Red | Green | Green | Green | Red | Green | Green | Red | Green | Green | Red | Green | Green | Red | Green | Red | Red | Green | Red | Red |
| Management of complications of labour | Green | Green | NA | Green | Green | NA | Green | Green | NA | Green | Green | NA | Green | Green | NA | Green | Green | NA | Green | Green | NA |

[1] Green = Available, Red = Not available, N/A = Not applicable

However, according to the Newborn Essential Solution and Technologies (NEST360˚) programme's recommendations, staff is advised on how to prevent infection in such unavoidable circumstances by grouping babies with similar conditions together [29]. With the help of commonly available, affordable medical technologies combined with comprehensive, evidence-based newborn care, ongoing clinical and biomedical mentorship, and the use of data to guide decision-making, NEST360˚ intends to lower the rate of inpatient newborn death in African countries (Malawi, Kenya, Tanzania and Nigeria) [29].

We found that power and water supplies were often interrupted despite the adapted Malawi standards for improving quality of maternal and newborn care [21] emphasizing the importance of adequate water and energy supplies. Birth asphyxia, prematurity and respiratory distress are common in this setting [6] making power-dependent equipment such as oxygen concentrators, continuous positive airway pressure (CPAP) and phototherapy essential for reducing mortality. Inadequate infrastructure has been reported previously in Malawi [16, 17, 30] compromising compliance with the WHO 2020 standards on supplemental oxygen and CPAP management. Oxygen concentrators, CPAP, phototherapy, suction machines, and pulse oximeters are some of the technologies that NEST360˚ is using to equip staff on its use and care to support sick neonates [29]. However, much of this technology needs a constant power supply to operate, which is rarely constant in this context. In situations when the electrical grid is intermittent, the use of renewable energy systems has been encouraged [31]. Some Malawian facilities (including one in this study) received solar power packs from UNICEF or global funds in 2017–2018, however there are still issues with management and technical capability at the central and facility levels for the continuous operation and maintenance of solar systems [31].

## Transport and communication

Timely referral for obstetric emergencies is vital in preventing morbidity and mortality in the newborn. A critical shortage of ambulances was noted in these hospitals despite the adapted Malawi standards which stress having a pre-established plan for timely referrals. The hospitals were compliant with the adapted standards having either a mobile phone, landline or radio that functions all times. Reliable communication channels are crucial as theatre staff, specialists and medical officers are often not available during off-time hours in case of emergency. Referral to higher-level care and communication tools were among proposed signal functions for supporting quality obstetric and newborn care in a literature review and expert opinion survey with 39 international experts from LMICs [32]. Due to inadequate funds, district hospitals face challenges in maintaining or fuelling the ambulance which may cause delay. However, a systematic review that included studies conducted from Malawi and Ghana found that use of motorcycle ambulances reduced referral delays [33]. Although a birth preparedness and complication readiness plan is encouraged for pregnant mothers, only 24% of pregnant women in the second and third trimester in an Ethiopian study were prepared for delivery and obstetric emergencies, that include emergency funds or transport in the event of a complication or onset of labour [34]. Birth preparedness and complication readiness need to be intensified during antenatal health education.

## Staff availability and training

Compared to the WHO-recommended threshold of 4.45 doctors, nurses, and midwives per 1000 population [35], the Malawi health care system is severely understaffed with 0.5 clinical and nursing staff per 1000 population [36], nine times less than recommended threshold. All hospitals had a critical shortage of registered nurses/midwives as the proportion of filled posts

were below 50%. At the same time, hospitals 1 and 6 also had a critical shortage of medical officers/specialists [36]. In comparison to other central hospitals in Malawi, hospital 1 is severely understaffed, with a vacancy rate of 51% for clinical staff and 58% for nursing/midwifery staff [36]. This understaffing resulted in only nursing/midwifery and support staff always available during day and night shifts. Studies conducted in Bangladesh and Malawi reported that numbers of nursing/midwifery staff were insufficient with staff facing excessive workloads that surpass their capacity to cope during the night shift, compromising the quality of care [37]. Clinicians were not always available during the night shift across all three areas compromising the required skill mix to manage sick newborns and putting an extra workload on the night duty nurse. Only one to two clinicians were available on call to cover both the paediatric and neonatal units for hospital 1 or to cover the whole hospital for hospitals 2–7. These findings are at odds with the adapted Malawi standards on having competent, motivated staff consistently available to provide routine care and manage complications.

Furthermore, we found a lack of specialised paediatricians/neonatologists and few trained clinical officers in obstetrics /gynaecology and paediatrics. In an attempt to compensate for the long-standing shortage of skilled staff [38], Malawi has adopted the use of mid-level cadres like clinical officers, medical assistants, and nurse-midwives at registered, enrolled and technician grades to provide both emergency and routine care [37, 39, 40]. Recently, Malawi introduced a two-year speciality training programme for qualified clinical officers in obstetrics /gynaecology, paediatric, surgery and internal medicine. This should increase the number of trained clinical officers in obstetric and paediatric specialities to ensure that at least one clinical officer, well equipped with maternal and neonatal care skills, is always available during the day, night, and weekend shifts.

Although mid-level cadres can help reduce stillbirths and neonatal deaths in LMICs, these staff require in-service training to update skills and competencies [41–43]. Nurses in LMICs obtain competence in neonatal care through training on the job [44]. In the absence of speciality training in Malawi, some nurses and clinicians working in the neonatal unit had benefitted from occasional in-service training such as COIN. However, we found inadequate training in all hospitals for HBB, COIN, IMNC and maternal and neonatal death audits. This is inconsistent with the adapted Malawi standards that recommend regular in-service or refresher training every 12 months. As training alone is not effective in improving quality of care, we recommend the inclusion of supportive supervision from both national and district levels as suggested by Rowe and colleagues [45, 46].

## Availability of essential supplies, drugs and equipment

Material resources are vital to providing quality care during childbirth with a better user experience [15]. Despite the adapted Malawi standards and like other LMICs [16, 17, 27, 30], we found that facilities were underequipped with many essential drugs, supplies, equipment, and laboratory items. Interestingly, despite all hospitals procuring supplies and drugs from the same central medical stores, we observed variations in stocks of essential drugs and supplies between hospitals. For example, one hospital had 8 out of 21 items out of stock in January, while others had only 1–4 items out of stock. This suggests a need to improve drug and supplies needs assessment to ensure continuous availability of items at the hospital level with an adequate budget allocated. A review of case studies of interventions that generated or maximized resources to facilitate effective public health interventions in three LMICs (Zambia, Zimbabwe and Madagascar), recommended that evaluating and identifying contextual factors that influence the feasibility of interventions should be a top priority for researchers and implementers [47]. The contextual factors influencing the feasibility of these interventions

included leadership engagement, local capacity building, infrastructural support for multilevel scale up and cultural and contextual adaptation [47].

## Clinical protocols

The adapted Malawi standards emphasise the need for written, up to date clinical protocols. These should be consistent with WHO guidelines [19] and address routine maternal and newborn care, complicated pregnancy and labour, preterm labour, infection prevention, care of small and preterm babies, resuscitation of babies who cannot breathe and essential newborn care. However, clinical protocols were missing in some hospitals. A study in Ethiopia [48] and a systematic review for LMICs [49] found unavailability of protocols on essential newborn care and neonatal resuscitation in hospitals. In an Ethiopian study that surveyed 741 health facilities, of which 14% were hospitals and 86% were health centres, only 60% and 37% of essential newborn care guidelines were available in hospitals and health centres, respectively [48]. Four neonatal intensive care units in Thailand lacked dissemination of practice protocols to nurses and Ethiopian hospitals and health centres that conduct deliveries had no guidelines and protocols on essential newborn care and neonatal resuscitation [49]. Similarly, in this study, labour ward and postnatal wards mostly lacked newborn care guidelines.

## Leadership and governance

Good managerial and clinical leadership improve performance by directing staff and creating an environment for support [19]. Supportive supervision and performance appraisal, identified as a gap in this study, accompanying the provision of resources are integral to improving health care, worker job satisfaction, motivation, and performance. But supervisors in LMICs often lack skills, tools, and transportation, are overburdened with administrative duties, and wait for a financial incentive (per diem). As a result, supervision visits are missed with little accountability as to whether supervision is done or not [46]. Despite the adapted Malawi standards which advocate for competent, motivated staff consistently available to provide routine care and manage complications, we found that district management teams failed to supervise their own facilities. Only the national level team visits facilities on quarterly basis and gives feedback. It is a requirement for national level (MoH) staff in Malawi to supervise districts, similarly the DHMT members supervise their facilities on quarterly basis [50]. The DHMT acts as both external supervisors for health centres as well as internal supervisors within their own departments and wards, usually using a checklist and giving feedback [50]. In a study conducted in 9 hospitals and 45 health centres in Malawi, only 57% of staff felt that they were adequately supervised [51]. In Tanzania, barriers to supervision included lack of a clear policy, limitations in measuring quality improvement progress and resource constraints such as funds and limited number of supervisors [52].

Even though facility interventions may enhance care, facility leaders should strive to offer comprehensive care at all system levels, including in the community. In the Maikhanda programme in Malawi, more neonatal deaths were prevented when quality improvement at the facility level was implemented in combination with community involvement than when it was done alone [53].

## Guidelines and standards implementation

Despite internationally recognised WHO guidelines and standards, challenges have been reported on the operationalisation of guidelines for maternal health in LMICs [54]. WHO recommends the adaption of standards to suit the context of each country [19] and in Malawi, a multidisciplinary working group comprising clinicians, nurses, policymakers and

development partners adapted the WHO standards and, recognising the importance of community engagement, added a ninth standard on community health care and social accountability for maternal and neonatal health [55]. Despite these efforts at the national level, this study has revealed deficiencies in the support for delivering quality care during delivery and for newborns. WHO has provided an implementation approach for the standard with seven steps according to the 'Plan Do Study Act' model: establishing leadership structures and functions, adapting standards of care, conducting a baseline situation analysis or assessment, ensuring essential infrastructure to get started, building capability and implementing interventions, monitoring progress, and refining the strategy [19]. This study has reported deficiencies that provide a basis for developing interventions to improve standard implementation in Malawi. In addition, the WHO and Malawi adapted standards on maternal and newborn care lack clear monitoring and evaluation plans and tools to improve the implementation guidance and learning platform [19, 21].

We also observed variations among hospitals in terms of the management of necessary medications and supplies. This could be because the hospitals typically execute an incomplete drug assessment plan before placing orders from the central medical store, In a qualitative study which examined the facilitators and barriers to implementing stillbirth and neonatal death audit [56], causes of insufficient supplies included limited facility autonomy and decision-making powers despite decentralisation, one sole supplier for drugs and essential drugs with limited powers to outsource if they were not available, and inadequate leadership support. Financial support also remained inadequate in the facilities, which depended on external donors to fund activities, resources and equipment. This brings variations as donors operate within their specified catchment region resulting in some hospitals receiving more support than others.

## Strengths and limitations

Strengths of this study were the inclusion of one central and six district hospitals from 7 districts in Malawi increasing the applicability of the findings. We also used a WHO validated checklist (SARA) for assessing service availability and readiness making comparisons with other studies easier. This tool was also used for the Maikhanda quality improvement project evaluation in Malawi to assess the facility resources [30]. The assessed parameters were comprehensive, including maternal and newborn services, physical infrastructure, availability of resources, equipment and supplies, guidelines, staffing, training, leadership and governance. The methods generated a wealth of information to identify gaps and recommend improvement.

This assessment also had several limitations. It was only conducted in the southern region of Malawi, which may limit generalisation at the national level. Despite decentralisation, the management of human, material, financial, and donor resources is centralised across three regions in Malawi. The insights from this study may be useful to the other regions in Malawi and LMICs with similar management systems. The survey did not capture caesarean section capacity as an important approach to reducing stillbirths and early neonatal deaths. The survey only reported general readiness needed to prevent stillbirth and neonatal deaths, not necessarily differentiating readiness to prevent stillbirths and where that overlaps and differ for neonatal deaths in hospitals and post discharge. Information was primarily from self-reports; further studies could include direct observation of care to confirm data reliability and semi-structured interviews with staff and women to understand their experience of care (standards 4–6). Due to financial and time constraints, our assessment parameters did not include the following components of the maternal and newborn care standards, actionable information system,

effective communication and family participation, respective women and newborn care and emotional and psychological and developmental support. Also, we did not assess the referral feedback system. Finally, our study did not include observation of care delivery, community components of care and quality improvement that could have provided additional information on facility readiness. Our assessment focused on in hospital care to prevent stillbirths and neonatal death but little on prehospital care and post-discharge care. However, since most of our findings were consistent across the seven facilities, we consider that the study provides information that can guide interventions, implementers, policy, and researchers to improve the quality of care for newborns and outcomes.

## Conclusion

Human and material resources to provide quality care during delivery and for newborns were mostly inadequate and inconsistent with the Malawi standards. Assessing the current status of resource availability and barriers to delivering care has highlighted gaps in the system. These provide a basis for health care professionals, policymakers, health service planners, programme managers, regulators, professional bodies and technical partners involved in maternal and newborn care to help in planning, delivering and ensuring the quality of health service delivery. Addressing these deficiencies would be expected to lead to better newborn outcomes. A multi-country evaluation study is needed to better understand and identify ways of mitigating challenges in the implementation of WHO or adapted quality standards.

## Supporting information

**S1 Checklist. Inclusivity in global health.**
(DOCX)

**S1 Appendix. Health facility resource survey tool.**
(DOCX)

**S1 Table. Standards of care and quality statements.**
(DOCX)

**S2 Table. Total ambulances by facility.**
(DOCX)

**S3 Table. Training of clinicians and nurses.**
(DOCX)

**S4 Table. Number of stockout days in the preceding month of essential supplies and drugs in pharmacy.**
(DOCX)

## Acknowledgments

The authors would like to thank staff and management team members from all seven participating hospitals who provided this study information. We would also like to thank Ismaela Abubakar from the Clinical Trial Unit at LSTM, who helped with the development of forms and tables in Microsoft Access. Thanks to Dr Joana Raven and Dr Florence Mgawadere from LSTM for guidance during concept development and manuscript review. Many thanks to Professor David Lissauer and Maternal Health group and Mtundu khongono from Malawi-Liverpool Wellcome Trust, Clinical Research Programme, for support during fieldwork.

## Author Contributions

**Conceptualization:** Mtisunge Joshua Gondwe.

**Data curation:** Mtisunge Joshua Gondwe.

**Formal analysis:** Mtisunge Joshua Gondwe.

**Funding acquisition:** Mtisunge Joshua Gondwe.

**Investigation:** Mtisunge Joshua Gondwe.

**Methodology:** Mtisunge Joshua Gondwe.

**Project administration:** Mtisunge Joshua Gondwe.

**Resources:** Mtisunge Joshua Gondwe.

**Software:** Mtisunge Joshua Gondwe.

**Supervision:** Nicola Desmond, Mamuda Aminu, Stephen Allen.

**Validation:** Nicola Desmond, Mamuda Aminu, Stephen Allen.

**Visualization:** Nicola Desmond, Mamuda Aminu, Stephen Allen.

**Writing – original draft:** Mtisunge Joshua Gondwe.

**Writing – review & editing:** Nicola Desmond, Mamuda Aminu, Stephen Allen.

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
