## [Decision Letter · Decision Letter 0]

10 Aug 2022

PGPH-D-22-00596

Resource availability and barriers to delivering quality care for newborns in health facilities in the southern region of Malawi: a multisite observational study

Dear Dr. Gondwe,

Thank you for submitting your manuscript to PLOS Global Public Health. After careful consideration, we feel that it has merit but does not fully meet PLOS Global Public Health’s publication criteria as it currently stands. Therefore, we invite you to submit a revised version of the manuscript that addresses the points raised during the review process.

Please note that we have only been able to secure reviewers to assess your manuscript. We are issuing a decision on your manuscript at this point to prevent further delays in the evaluation of your manuscript. Please be aware that the editor who handles your revised manuscript might find it necessary to invite additional reviewers to assess this work once the revised manuscript is submitted. However, we will aim to proceed on the basis of these reviews if possible. 

We look forward to receiving your revised manuscript.

Kind regards,

Julia Robinson

Executive Editor

Journal Requirements:

Additional Editor Comments (if provided):

Reviewers' comments:

Reviewer's Responses to Questions

**Comments to the Author**

1. Does this manuscript meet PLOS Global Public Health’s publication criteria? Is the manuscript technically sound, and do the data support the conclusions? The manuscript must describe methodologically and ethically rigorous research with conclusions that are appropriately drawn based on the data presented.

Reviewer #1: Yes

Reviewer #2: Partly

2. Has the statistical analysis been performed appropriately and rigorously?

Reviewer #1: Yes

Reviewer #2: Yes

3. Have the authors made all data underlying the findings in their manuscript fully available (please refer to the Data Availability Statement at the start of the manuscript PDF file)?

Reviewer #1: Yes

Reviewer #2: Yes

4. Is the manuscript presented in an intelligible fashion and written in standard English?

Reviewer #1: Yes

Reviewer #2: Yes

5. Review Comments to the Author

Reviewer #1: Thank you for the opportunity to review this manuscript. It is very well written and organized. The topic area is important and practical. It is also very timely with the recent publication and dissemination of the new standards for sick and small newborn care as well as investment and interest in the health and survival of these vulnerable infants. This paper captures important information on the transition from policy to practice. I have a number of suggestions below that I think could strengthen this paper and highlight more concretely its added value. Thank you for conducting and writing up this work.

INTRODUCTION

- It would be helpful to provide a bit more context on what other related work has been done in Malawi and what gap this study fills / what is the added value. A couple studies worth looking at and discussing if you have not already: Kwaza 2020 BMC Health Services Research, Leslie 2016 PLoS, etc.

RESULTS

- Summary: In the summary you note that an ambulance is available as a positive but throughout this is noted as a barrier. I suggest that you move this to the second paragraph of the discussion as refer to it as a barrier.

- I think it is important to mention that you have not looked into all aspects of the standards and why. For example, you do not discuss health information systems, demand side pieces, respectful care, etc.

DISCUSSION

- Infrastructure: You say that there are inconsistencies with the WHO standards. Can you please elaborate a bit more on what those differences are to help the reader connect the standards with your data without having to reference the standards themselves? It would be good to have a short section summarizing the key standards relevant to this work somewhere in the paper and/or referencing them in a figure/table together or separate from the information gathered as part of this study.

- Transport and communication: You write “The hospitals were compliant with the adapted standards having either a mobile phone, 263 landline or radio that functions all times.” However, in the results, you noted that “All hospitals had at least one functional landline telephone or mobile 165 phone at all times except hospitals 1 and 6, where the mobile phone was not always 166 available for use.” Please make sure these are consistent.

- Transport and communication: The last sentence – in the WHO standards? Where?

- How are these facilities generalizable to others in different regions in Malawi?

- Can you please explain some of the potential reasons behind the differences between health facilities?

- What about financial aspects and what role they plan in all of this?

Reviewer #2: The authors have written a concise description of the readiness among a sample of hospitals in southern Malawi to provide essential newborn care. Overall, the manuscript is clearly written and the study is robust in having used a standard tool adapted to Malawi to measure readiness, one which apparently was used in other related QI work in Malawi. The team has used some strong data visualizations to synthesize a range of data, however there are areas where a reader is still left not quite sure of the results. The paper is important in further adding to the evidence of the need to translate policy into resources needed as a first step to ensuring quality and equitable care. As such, I had a number of suggestions which I believe would strengthen the manuscript and its value to the broader community working to improve outcomes for newborns and reduce still births

Introduction

The authors describe the work by WHO establishing standards and a process for improvement. For someone less fluent, it would be helpful to use a few sentences to detail the guidance and the 7 steps. This will also help interpret the results in the broader context. Similarly, any existing data on hospital (and health center) readiness from Malawi would frame this study, as well as patterns of care including FBD, post-natal care and any insights into timing and causes of neonatal deaths in Malawi (ex. In-hospital versus post-discharge) as well as any data from the audits described. They also describe an additional area of CHWs, but this is not assessed nor discussed.

Methods:

It was interesting that the survey was done during COVID-19-what if any impact and /or precautions were taken?

It was also a little unclear why these were chosen and why HCs (who likely encounter significant stillbirth and need to transfer sick young infants). This also needs to be in the limitations

Did the survey also capture C-section capacity as an important approach to reduce stillbirths? What were the components for managing sick- young infants (I saw CPAP, KMC support? Feeding?

Results

The results are details and a bit more information would be helpful for a reader. Many of the results are described qualitative, requiring looking at the table or appendix. Adding in some numbers/range to the text would be helpful (ex. Essential supplies). Similarly, when the term “sometimes” is used, need to define in methods what that means (anything between none and always?)-particularly for Table 2

In table 1-some measure of volume would help understand the workload by hospital. It would also be helpful to have the number as well as % for positions filled (and was this overall versus larger or smaller gaps in some areas?

Why were only 3 district hospitals allocated ambulances? This should be discussed in the discussion. Also, I did not see anything about functioning of the referral system (both to and from the hospitals) beyond ambulance and phone availability. Were there metrics in the adapted survey which asked about things like counter referrals or feedback?

Similarly-since hospital 6 combined their post-natal ward-I was confused why authors did not assess the combined ward for both competencies and staffing since PNC was being delivered

Did any of the hospitals have a specific area for sick and LBW infants (even if not officially a NICU?

Table 2 is a nice way to visualize data. Is that any versus none?

The training gaps are well noted, but the standard is not discussed until in discussion. It would be helpful to have in an appendix (or noted in the text) what is the standard, so a reader knows in results where there are gaps. IN the discussion As the authors point out, there were gaps in required CPD (although also limited if any evidence on impact), What dd the MOH supervisors do?

For supplies, Why is oxytocin needed in neonatal ward? Was there no HCT but full blood count? Was bilirubin available through sending out?

Were there any summary scores for the areas measured and specifically for care of sick young infants (some emerging approaches for this)

Discussion

The authors have a nice description of the findings, but I would like to see a bit more digging into why and possible solutions from the growing work and literature to improve neonatal care in Malawi (ex. Makonde, NEST) and the region.

Some specific comments:

P20 they note that clinicians were not always available during night shift-was this across all 3 areas? What was their workload?

See comment above about CPD. The sentence calling for more studies of CPD on outcomes seems odd given the extensive evidence that training and retraining alone is not effective but must include supportive supervision (see Rowe)

On page 22, they have a bit if a value comparison with Ethiopia and a systematic review -It would be good to have a bit more granularity-what was similar and different at the hospital level? Any insights into variability between the surveyed hospitals (this does need more discussion)

The section on p22 on supervision is a nice background on why it is important, but no discussion on what the MOH does, and what the findings in Malawi compare with other published work

I was curious the reasoning behind a larger survey needed, versus one with plans and resources to address the gaps. I was also interested in who the community components could be measured (as I think it was not in the modified SARA)

I would add to the limitations that observation/delivery of care was not included (see Das et al for the gap between inputs and delivery) and no data on the community components nor on the QI. I think also missing is How to understand the readiness needed to prevent still birth and where that overlaps and differs for neonatal (in hospital and post discharge)

6. PLOS authors have the option to publish the peer review history of their article (what does this mean?). If published, this will include your full peer review and any attached files.

**Do you want your identity to be public for this peer review?** For information about this choice, including consent withdrawal, please see our Privacy Policy.

Reviewer #1: No

Reviewer #2: No

---

## [Decision Letter · Decision Letter 1]

8 Nov 2022

Resource availability and barriers to delivering quality care for newborns in health facilities in the southern region of Malawi: a multisite observational study

PGPH-D-22-00596R1

Dear Mrs Gondwe,

We are pleased to inform you that your manuscript 'Resource availability and barriers to delivering quality care for newborns in health facilities in the southern region of Malawi: a multisite observational study' has been provisionally accepted for publication in PLOS Global Public Health.

Best regards,

Julia Robinson

Executive Editor

Reviewer Comments (if any, and for reference):

Reviewer's Responses to Questions

**Comments to the Author**

1. If the authors have adequately addressed your comments raised in a previous round of review and you feel that this manuscript is now acceptable for publication, you may indicate that here to bypass the “Comments to the Author” section, enter your conflict of interest statement in the “Confidential to Editor” section, and submit your "Accept" recommendation.

Reviewer #2: All comments have been addressed

2. Does this manuscript meet PLOS Global Public Health’s publication criteria? Is the manuscript technically sound, and do the data support the conclusions? The manuscript must describe methodologically and ethically rigorous research with conclusions that are appropriately drawn based on the data presented.

Reviewer #2: Yes

3. Has the statistical analysis been performed appropriately and rigorously?

Reviewer #2: Yes

4. Have the authors made all data underlying the findings in their manuscript fully available (please refer to the Data Availability Statement at the start of the manuscript PDF file)?

Reviewer #2: Yes

5. Is the manuscript presented in an intelligible fashion and written in standard English?

Reviewer #2: Yes

6. Review Comments to the Author

Reviewer #2: The authors have done a through job in responding to comments which has significantly improved the paper. My only minor suggestion is making clear the level of facilities this paper is addressing in the title (hospital) as this does not include health centers

7. PLOS authors have the option to publish the peer review history of their article (what does this mean?). If published, this will include your full peer review and any attached files.

**Do you want your identity to be public for this peer review?** For information about this choice, including consent withdrawal, please see our Privacy Policy.

Reviewer #2: No
